# Adaptive Quantization Error Reconstruction for LLMs with Mixed Precision

**Lin Ou,**[*] **Jinpeng Xia,**[*] **Yuewei Zhang, Chuzhan Hao & Hao Wang**[†]
Alibaba Cloud, Alibaba Group
{oulin.ol,qihao.xjp,liyou.zyw,haochuzhan.hcz}@alibaba-inc.com
cashenry@126.com

## Abstract

Large language models (LLMs) has demonstrated superior performance on various downstream tasks. However, their practical applications are hindered by their immense memory and computation requirements. Although recent post-training quantization methods can effectively reduce memory usage and improve computational efficiency, they often overlook the varying sensitivity of different layer weights to bit precision. Additionally, the previous methods suffer from significant accuracy loss under low-bit quantization (2-3 bits). To address these limitations, we propose Adaptive Mixed Precision and Low-Rank Quantization Error Reconstruction for LLMs (AMLQ), which achieves state-of-the-art performance under the approximate average bit precision overall. Furthermore, we introduce the low-rank decomposition to reconstruct quantization error based on the output features. Experimental results demonstrate that this method can be effectively combined with various quantization techniques and bring considerable performance gains. Our approach comprehensively considers model performance and inference efficiency, offering more than $3\times$ speedup over the FP16 execution.

## 1 Introduction

Large language models (LLMs) such as GPT-4 (Bubeck et al., 2023) and Qwen (Bai et al., 2023), have shown excellent performance on various natural language processing (NLP) tasks (Brown et al., 2020; Touvron et al., 2023a; Zhao et al., 2023; Feng et al., 2024). Meanwhile, experimental evidence indicates that *emergent capabilities* only manifest when the model scale is sufficiently large (typically exceeding 10B parameters) (Wei et al., 2022a). This also results in considerable computational and memory requirements, posing significant challenges for the practical applications in real-world scenarios. Therefore, quantization techniques capable of mitigating computational and memory pressures exhibit excellent prospects in the era of LLMs (Frantar et al., 2022; Dettmers et al., 2023; Xiao et al., 2023).

Quantization is often categorized into two main approaches: post-training quantization (PTQ) and quantization-aware training (QAT). While QAT tends to maintain better experimental accuracy than PTQ, its high training cost renders it less practical. Therefore, PTQ becomes an attractive research task for reducing computational and memory cost (Dettmers et al., 2023; Lee et al., 2023a; Shao et al., 2023; Lin et al., 2024). PTQ is a technique that quantizes a pre-trained LLM directly, without additional training. The quantization error propagates and accumulates through the LLMs, leading to substantial task accuracy degradation. To maintain the original model accuracy, Frantar et al. (2022) employs second-order information to iteratively round grouped weights and correct the quantization error. Additionally, due to the presence of magnitude outliers in the model layer weights, the quantization process is severely affected (Wei et al., 2022b; Xiao et al., 2023). Thus, an intuitive approach involves identifying and isolating outliers within the weight distribution.

---

[*] The first two authors contributed equally
[†] Corresponding author

These outliers can then be subjected to mixed-precision quantization or weight scaling. Specifically, certain salient weights can be retained in FP16 (half floating point) at finer granularity while quantizing the rest to low-bit precision (Dettmers et al., 2023; Tang et al., 2024). Alternatively, protecting the salient weights can be implemented by learning fine-grained weight scaling (Lee et al., 2023b; Luo et al., 2023; Lin et al., 2024; Zhang et al., 2024). Chee et al. (2024); Egiazarian et al. (2024) introduce codebook mechanism to achieve low-bit quantization (2 to 3 bits per parameter).

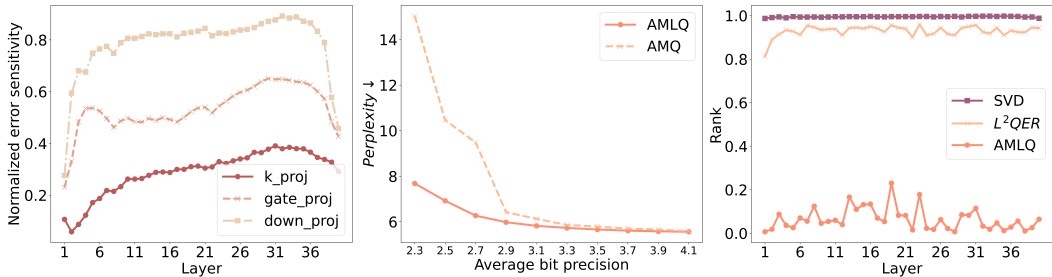

Figure 1: Motivation of our proposed AMLQ. (a) The varying quantization error sensitivity of different layer weights in LLMs. We observe that the deep layers are more sensitive than others in quantization bit precision. (b) The introduction of the low-rank approximation quantization error reconstruction can mitigate the significant degradation of model performance at low-bit precision. (c) The output activation errors decomposed by our approach are low-rank, while the layer weight errors are not low-rank.

However, empirical evidence suggests that different layers in a neural network have the varying importance or redundancy (Han et al., 2015; Liu et al., 2019). Similarly, different layer weights in LLMs exhibit varying sensitivity to bit precision, as illustrated in Figure 1. Therefore, global quantization strategies often do not achieve optimal experimental results. We expect to get the optimal bit precision combination for each layer weight. In addition, the previous methods Frantar et al. (2022); Dettmers et al. (2023); Lin et al. (2024) suffer from significant accuracy loss under low-bit quantization (Figure 1). Although using the codebook can have good performance (Egiazarian et al., 2024), it is relatively slow in inference. Zhang et al. (2024) also leverages low-rank approximation to compensate weight errors of low-bit quantization to some extent. However, Yu & Wu (2023) proves that the weights of transformer-based models are surprisingly not low-rank, which makes it less effective to leverage low-rank approximation to correct quantization errors base on layer weights. Simultaneously, this low-rank error reconstruction method is also sensitive to different layer weights.

In this paper, we address these limitations and propose adaptive mixed precision and low-rank quantization error reconstruction for LLMs, dubbed AMLQ. The core idea of our method is to adaptively select bit-precision and the rank of the error correction matrix for different layer weights. First, we propose an activation-induced adaptive search method for selecting optimal bit precision for different layer weights. Through this approach, we achieve a relatively optimal parameter setting under a predefined average bit precision. Second, to further correct quantization errors, we also introduce an adaptive low-rank quantization error reconstruction based on activation. Compared to low-rank approximation applied directly to model weights, this method demonstrates superior performance. Overall, we keep higher-bit precision and higher-rank correction matrices for more sensitive layers, while adopting more aggressive compression strategies for less sensitive ones. This error reconstruction method also has the flexibility to adapt to other advanced quantization techniques, providing plug-and-play capabilities. Our contributions are as follows:

- We propose an adaptive mixed precision and low-rank approximation quantization error reconstruction, which allows us to automatically search for optimal parameter settings in a discrete space.

- Experimental results demonstrate that our approach achieves state-of-the-art performance at the approximate average bit precision. In the case of low-bit quantization, our

approach also reaches relatively fast inference speed. Simultaneously, our proposed low-rank quantization error reconstruction can be plugged into other advanced quantization techniques to optimize their performance.

## 2 Related work

**Post-Training Quantization of LLMs.** The PTQ is a very practical technique for the application of LLMs. It remains challenging due to the presence of magnitude outliers in model weights and activations. Exiting methods can be broadly categorized into two fields: weight-only and weight-activation quantizations. Weight-only quantization focuses on converting weights to low-bit precision. Frantar et al. (2022) implements block-wise iterative quantization error correction using grouped weights. Dettmers et al. (2023); Lee et al. (2023a); Lin et al. (2024) emphasize that a small fraction of salient weights are much more important for LLMs' performance compared to others. Therefore, Dettmers et al. (2023) and Lee et al. (2023a) employ mixed-precision quantization to safeguard vital weights, while Lin et al. (2024) opts for channel-wise scaling to protect salient weights induced by activations. Weight-activation quantization compresses both weights and activations, which can provide computational improvements, but execute with large amounts of accuracy loss relative to their uncompressed counterparts (Yuan et al., 2023; Shao et al., 2023). In this paper, we focus on the weight-only quantization because we expect to keep the model performance as much as possible after quantizing.

**Mixed-Precision Quantization.** Mixed-precision quantization refers to the process of assigning appropriate quantization bit-widths to the weights and activations of each layer in a model. The goal is to achieve an optimal balance between accuracy and hardware metrics in the quantized model. Dong et al. (2019) utilizes second-order information to assess the quantization sensitivity of each layer in the model, thus allocating the appropriate bit-width. Wang et al. (2019) employs reinforcement learning techniques to search for the quantization bit-widths of different layers, which incorporates feedback from hardware simulators and achieves a hardware-aware mixed-precision strategy. Additionally, there are some methods to implement this using training-based algorithms (Cai & Vasconcelos, 2020; Yang & Jin, 2021). However, the above methods are not yet easily adapted to quantization for LLMs.

**Low-Rank Quantization Error Reconstruction.** LoRA (Hu et al., 2022) leverages the concept of low-rank matrices to make the training process of LLMs extremely efficient and fast, which can significantly reduce the trainable parameters and GPU memory. Dettmers et al. (2024); Li et al. (2024) introduce quantification techniques into LoRA-based tuning methods, which are not post-training quantization method and fuse the low-rank matrices back to original high-bit weights. Zhang et al. (2024) leverages the separated low-rank matrices to correct the quantization weight errors. However, it ignores the fact that the quantization weight error matrix is high-rank, even after scaling it.

## 3 Methodology

Our method requires a small set of calibration data to measure the sensitivity of different layers and the importance of intra-layer channels, and then adaptively selects the bit precision of layer weights and the rank of error reconstruction matrices. Our method works on all linear layers in LLMs except the weights of embedding and LM head. According to the weight of the i-th layer $W \in \mathbb{R}^{m*c}$, the input activation value $X \in \mathbb{R}^{c*n}$, and the quantized weight $W_q$, the final quantitative target can be denoted as:

$$\min \|WX - W_qX\|_2 \tag{1}$$

### 3.1 Mixed-precision quantization

As mentioned above, the quantization sensitivities vary among layers and intra-layer channels. Therefore, to achieve the predefined average bit precision (also known as BPW, bit per weight) for the entire model, different quantization bit-widths can be used for different

layers or channels during quantization. The sensitivity $\delta$ is measured by calculating the output loss after quantization, which is computed as follows:

$$\delta = \|WX - W_q X\|_2^2 / \|WX\|_2^2 \tag{2}$$

where $W \in \mathbb{R}^{m*c}$ is the weight of the i-th layer, $X \in \mathbb{R}^{c*n}$ is the activation of the input value, $W_q$ is the quantized weight. $\|\cdot\|_2$ represents the function of $\ell^2$-norm. For the weight of each layer, we will first define our quantization bit-precision candidate set $B \in \{2, 3, 4, 5, 6, 8, 16\}$, and then preset different proportions of mixed-precision combinations from the $B$ set, with a total of $n$ possibilities. All combinations form a discrete search space $\mathcal{Q} \in \{q_1, q_2, \cdots, q_n\}$. For each combination, we can calculate the quantization loss $l_i(q_j)$ and the corresponding total bit size $b_i(q_j)$ for the $i$-th layer weights after quantization of the $j$-th mixed-precision combination. Finally, our goal is to minimize the objective function $L$ as follows:

$$\min_{\{q_j\}_{j=1}^n} L = \sum_{i=1}^l l_i(q_j) \tag{3}$$

$$s.t. \sum_{i=1}^l b_i(q_j) < P_{target\_bit} \tag{4}$$

where $l$ denotes the number of layers of the LLM. $P_{target\_bit}$ is the predefined upper bound on the number of bit parameters for the all model weights. This optimization process can be viewed as an integer programming problem, and it will be very time-consuming to find the exact global optimal solution from the whole search space $\mathcal{Q}$. The sensitivity of each layer in the model is independent of the quantization bit precision of the other layers. Therefore, we use a greedy strategy to approximate the solution. Although our greedy strategy may not always find the optimal configuration, it can greatly shorten the search time. In section 4, the experimental results also demonstrate that the mixed precision combination found through this greedy strategy can achieve state-of-the-art performance.

Specifically, we obtain the quantized weight $W_q$ in the following process. We comprehensively refer to the current superior GPTQ (Frantar et al., 2022) and AWQ (Lin et al., 2024) methods. First, AWQ proposes to use the magnitude of the input activation value to help scale the salient channel weights thereby protecting the salient weights during quantization. We use this method to calculate the average amplitude of activation $S_x = mean\_cout(X)$, and set a searchable hyperparameter $\alpha$ to control the intensity of the amplitude. Hence, the scaled weights are $W_s = S_x^\alpha * W$. To ensure that the output activation $Y$ is not affected, the input activation $X$ also needs to be scaled to $X_s = (S_x^\alpha)^{-1} * X$. The complete output activation is represented as follows:

$$Y = WX = W_s X_s = (W * S_x^\alpha)((S_x^\alpha)^{-1} * X) \tag{5}$$

We then calculate the inverse Hessian matrix $H^{-1}$ of the scaled input activation value (refer to GPTQ). Due to our scaling of weights and activations, we do not need to compute $XX^T$ repeatedly for reducing the amount of computation. The complete computation process is shown in Equation 6.

$$\begin{aligned} H^{-1} &= (2X_s X_s^T + \lambda I)^{-1} \\ &= (2\text{diag}((S_x^\alpha)^{-1})XX^T\text{diag}((S_x^\alpha)^{-1})^T + \lambda I)^{-1} \end{aligned} \tag{6}$$

where $\text{diag}(\cdot)$ denotes turning the vector into a diagonal array. The $H^{-1}$ will be used to evaluate the importance of each channel within the weight. Similarly, we also group channels and then assess the importance of weights within each group. For each group of weights, we reorganize the order, placing the more critical groups at the forefront. Furthermore, we introduce the idea of mixed precision, quantizing the more important groups with higher bit precision in the predefined search space. Finally, we can refer to the pseudocode in Algorithm 1 to calculate the quantized weight $W_q$.

## 3.2 Quantization error reconstruction based on low-rank decomposition

For the quantized $W_q$, there is a quantization error $\Delta W$ with the original FP16 weight $W$, which can be obtained by:

$$\Delta W = W - W_q \tag{7}$$

The rank of $\Delta W$ is relatively high, as illustrated in Figure 1. There are significant errors to decompose it into a low-rank matrix using *Singular Value Decomposition* (SVD) directly. LQER (Zhang et al., 2024) introduces the scaling factor to reduce the difficulty of decomposition and achieves good results. Moreover, we observe that the layer weights are not low-rank but the output activation value is low-rank. Therefore, we directly decompose the output activation error $\Delta Y \in \mathbb{R}^{m*n}$ using *Principal Component Analysis* (PCA).

$$\Delta Y = WX - W_q X = \Delta W X \tag{8}$$

We regard $\Delta Y$ as $n$ instantiations of the random eigenvector $y$ (each in $\mathbb{R}^m$), and then compute the covariance matrix between them as follows:

$$\text{Cov}(y) = \text{E}[yy^T] - \text{E}[y]\text{E}[y]^T \tag{9}$$

where $\text{E}[\cdot]$ denotes the function of calculating expectation. Since $\text{Cov}(y)$ is a positive semi-definite matrix, its eigenvalue decomposition (i.e., PCA) is

$$\text{Cov}(y) = USU^T \tag{10}$$

We extract the first $k$ columns of $U \in \mathbb{R}^{m*m}$ and get $U_k \in \mathbb{R}^{m*k}, U_k U_k^T \approx I$, hence

$$y - \text{E}[y] \approx U_k U_k^T (y - \text{E}[y]) \tag{11}$$

$$y \approx U_k U_k^T y - U_k U_k^T \text{E}[y] + \text{E}[y] \tag{12}$$

Therefore, the quantization error $\Delta Y$ can be represented as

$$\Delta Y \approx U_k U_k^T \Delta W X - U_k U_k^T \text{E}[y] + \text{E}[y] \tag{13}$$

Let $U_k$ and $U_k^T \Delta W$ form two low-rank matrices $B \in \mathbb{R}^{m*k}$ and $A \in \mathbb{R}^{k*c}$ respectively, $\text{E}[y] - U_k U_k^T \text{E}[y]$ is superimposed on the bias $b$ of the original model. Therefore, the output feature $Y$ of a linear layer consists of the following parts:

$$Y = W_q X + BAX + b \tag{14}$$

The layer weight $W \in \mathbb{R}^{m*c}$ is usually a large matrix. For instance, in the case of Llama2-70B (Touvron et al., 2023a), the minimum dimensions for $\{m, c\}$ are $\{8192, 8192\}$. Additionally, the newly added two parameters $\{A, B\}$ will increase the average bit precision of $\frac{k*(m+c)*16}{m*c}$. When $k \approx 64$, it will only increase the average bit precision by less than 0.2. For the unbiased linear layer, our method adds $m$ additional parameters, which are only about 0.02% of the $W$ parameters, almost negligible.

As mentioned above, the quantization sensitivity of each layer is different. It can also be observed in Figure 1 that the rank of the quantization error of each layer is also varying. Therefore, we adopt the concept of adaptive search to determine the rank in the quantization error reconstruction process. The rank in low-rank decomposition is predefined in the set $r \in \{0, 32, 64, 96, 128\}$. Additionally, we combine it with our adaptive mixed-precision quantization. The whole optimization process is

$$\min_{\{q_j\}_{j=1}^n, \{r_k\}_{k=1}^5} L = \sum_{i=1}^l l_i(q_j, r_k) \tag{15}$$

$$s.t. \sum_{i=1}^l b_i(q_j, r_k) < P_{target\_bit} \tag{16}$$

where $l_i(q_j, r_k), b_i(q_j, r_k)$ denote the quantization error and the total number of bit parameters at the $i$-th layer choosing $k_j$ as rank and $q_j$ as quantization bit precision. $P_{target\_bit}$ can be obtained by multiplying the number of model parameters by the predefined average bit precision. We can get a sub-optimal solution in a short time using the same greedy strategy.

---

**Algorithm 1** AMLQ

---

1: **procedure** QUANTONELAYER($W, X, Groupsize, Qparams, Rank, Grid_\alpha$)
2:     $S_x = \text{average}(X, dim = 1)$
3:     **for** $\alpha, q_j, r_k$ **in** zip($Grid_\alpha, Qparams, Rank$) **do**
4:         $W_s = W * S_x^\alpha$
5:         $X_s = (S_x^\alpha)^{-1} * X$
6:         $H^{-1} = \textbf{Cholesky}((2X_s X_s^T + \lambda I)^{-1})^T$
7:         $W_s^R, g_{idx} = \text{reorder}(W_s, H^{-1})$
8:         $W_q = \text{reorder}(\text{quant}(W_s^R, Groupsize, q_i), g_{idx})$
9:         $\Delta Y = (W - W_q)X$
10:        $B, A, b = \text{decompose}(\Delta Y, r_k)$
11:        $\text{Quantloss} = \|WX - (W_q X + BAX + b)\|_2^2 / \|WX\|_2^2$
12:        $\text{Losslist.append}(\text{Quantloss}, \alpha, q_j, r_k)$
13:    **end for**
14:    **Return** Losslist
15: **end procedure**
16:
17: **procedure** QUANTMODEL($Layers, Groupsize, Qparams, Rank, P_{target\_bit}, Grid_\alpha$)
18:     **for** $W, X$ **in** $Layers$ **do**
19:         Losslist.extent(QuantOneLayer($W, X, Groupsize, Qparams, Rank, Grd_\alpha$))
20:     **end for**
21:     BestModel = **greedy_search**(Losslist, $P_{target\_bit}$)
22: **end procedure**

---

### 3.3 Inference performance

In the inference stage, we design a special CUDA operator for mixed-precision weights to perform high-performance inverse quantization and calculation processes. We also provide pseudocode in Algorithm 1. By reading the low-bit weights in one go using the INT4 vector type, the calculation intensity is increased, the memory access bottleneck is overcome, and the generation speed of the model decoding phase is improved. Compared to the FP16 model, the speedup can be more than 3 times. For the introduced low-rank error reconstruction module, we refer to Punica (Chen et al., 2023) to implement the bgmv operator, and the overall generation speed is only reduced by 5%.

## 4 Experiments

### 4.1 Experimental Settings

**Quantization**. We experiment with weight-only quantization. The default settings are INT4/INT3/INT2 per-channel weight quantization, with an average bit precision (or BPW) of about 2.3 to 4.3. Group-wise weight quantization is represented by 'g', whose sizes are set from {32, 64, 128} in our adaptive mixed precision method. We set the group size to 128 in other methods. Our grid size of $\alpha$ is set to 20 following Lin et al. (2024). The search space for the rank of the low-rank approximation matrices is set from {0, 32, 64, 96, 128}. When the rank of a layer is set to 0, this means that two matrices of low-rank decomposition are not added. Following Shao et al. (2023), our calibration dataset is sampled from the WikiText2 (Merity et al., 2017) dataset, which contains 128 samples of 2048 tokens.

**Baselines**. We conduct experiments on the currently most influential open-source LLMs of different scales, such as Llama (Touvron et al., 2023a), Llama-2 (Touvron et al., 2023b), and Qwen (Bai et al., 2023). We compare our approach with FP16 model, GPTQ (Frantar et al., 2022), AWQ (Lin et al., 2024), OmniQuant (Shao et al., 2023), QUIP# (Tseng et al., 2024), and LQER (Zhang et al., 2024) at the different BPW settings. OmniQuant and QuiP# are further optimized at low-bit precision quantization and QuiP# is the most effective quantization method at present. However, their actual inference efficiency is still low. Compared to

AMLQ, our proposed AMQ denotes the approach that uses only mixed precision and does not introduce low-rank error reconstruction.

**Evaluation**. Following previous methods (Frantar et al., 2022; Xiao et al., 2023; Lin et al., 2024; Zhang et al., 2024), we profiled the quantized models on language modeling tasks and downstream tasks. Specifically, we report the perplexity (PPL) on WikiText-2 (Merity et al., 2017) and the accuracy on ARC (easy) (Clark et al., 2018), ARC (challenge) (Clark et al., 2018), PIQA (Bisk et al., 2020), WinoGrande (Sakaguchi et al., 2021), HellaSwag (Zellers et al., 2019), MMLU (Hendrycks et al., 2021), CEval (Huang et al., 2023), GSM8K (Cobbe et al., 2021), and HumanEval (Chen et al., 2021).

## 4.2 Main Results

We firstly compare the performance of the quantization models by perplexity on the WikiText-2 dataset. The results of llama-7b, llama-13b, llama2-7b, llama2-13b, and llama2-70b quantized by AWQ, OmniQuant, Quip#, $L^2$QER, AMQ, and AMLQ respectively, are shown in Table 1. We also report the final average bit precision $w$ for each method. We can observe that the low-rank quantization error reconstruction module adds about 0.2 BPW at different bit precision. At almost all model sizes, AMLQ achieves better results than AWQ and OmniQuant. AWQ and OmniQuant deal with outliers heuristically, while LQER uses an error reconstruction method. We observe that AMQ shows a significant accuracy degradation at 2-bit precision, but with the introduction of the error reconstruction, AMLQ outperforms OmniQuant. Quip# achieves the best quantization performance by codebook. However, its inference efficiency is slow and it is difficult to be applied in practical applications. In contrast, our proposed AMLQ, empowered by the CUDA operator, can achieve about 6 times the inference speed of Quip#.

| Method | Q Config | LLaMA | | LLaMA-2 | | | Avg. $\Delta$ PPL ($\downarrow$) | Avg. $w$ bit |
|---|---|---|---|---|---|---|---|---|
| | | 7b | 13b | 7b | 13b | 70b | | |
| FP16 | - | 5.68 | 5.09 | 5.47 | 4.88 | 3.32 | - | 16 |
| AWQ | INT4, g128 | 5.81 | 5.20 | 5.62 | 4.97 | - | 0.12 | 4.1 |
| OmniQuant | INT4, g128 | 5.77 | **5.17** | 5.58 | **4.95** | 3.40 | 0.08 | 4.1 |
| QuiP# | INT4 | **5.76** | **5.17** | **5.56** | **4.95** | **3.38** | **0.07** | 4.0 |
| LQER | INT4, g128 | 5.89 | 5.20 | 5.58 | 4.96 | - | 0.12 | 4.3 |
| **AMQ** | Dynamic | 5.79 | 5.18 | 5.57 | 4.96 | 3.40 | 0.09 | 4.1 |
| **AMLQ** | Dynamic | 5.77 | **5.17** | **5.56** | **4.95** | 3.39 | 0.08 | 4.3 |
| AWQ | INT3, g128 | 6.46 | 5.51 | 6.24 | 5.32 | - | 0.60 | 3.1 |
| OmniQuant | INT3, g128 | 6.15 | 5.44 | 6.03 | 5.28 | 3.78 | 0.44 | 3.1 |
| QuiP# | INT3 | **5.98** | **5.31** | **5.79** | **5.10** | **3.56** | **0.26** | 3.0 |
| **AMQ** | Dynamic | 6.13 | 5.40 | 6.01 | 5.20 | 3.66 | 0.38 | 3.1 |
| **AMLQ** | Dynamic | 6.04 | 5.38 | 5.82 | 5.13 | 3.64 | 0.33 | 3.3 |
| AWQ | INT2, g128 | 2.6e5 | 2.8e5 | 2.2e5 | 1.2e5 | - | 2.2e5 | 2.1 |
| OmniQuant | INT2, g128 | 9.72 | 7.93 | 11.06 | 8.26 | 6.55 | 3.81 | 2.1 |
| QuiP# | INT2 | **6.86** | **5.97** | **6.66** | **5.74** | **4.16** | **0.99** | 2.0 |
| **AMQ** | Dynamic | 15.19 | 8.73 | 15.03 | 8.68 | 8.53 | 6.34 | 2.3 |
| **AMLQ** | Dynamic | 7.81 | 6.49 | 7.68 | 6.34 | 4.64 | 1.70 | 2.5 |

Table 1: A comparison of perplexity ($\downarrow$) on WikiText-2. The highest value per column group is in **bold** and the second highest value is underlined. Partial results are from their original papers. AMLQ outperforms AWQ, OmniQuant, LQER, and AMQ. Most methods suffer significant performance degradation at low-bit quantization, but our approach is on par with QUIP# which is specifically optimized for the low-bit setting, while providing substantially higher inference efficiency.

| Size | Method | WinoGrande | PiQA | HellaSwag | ArcE | ArcC | Avg. accuracy (↑) |
|------|--------|------------|------|-----------|------|------|-------------------|
| 13B | FP16 | 72.14 | 79.16 | 60.06 | 79.46 | 48.38 | 67.84 |
| 13B | Quip# | 72.61 | 79.05 | 59.47 | 78.62 | 46.59 | 67.10 |
| 13B | GPTQ | 72.14 | 78.24 | 59.50 | 78.54 | 47.10 | 67.10 |
| 13B | AWQ | **72.69** | 78.67 | **60.12** | 78.91 | 47.10 | 67.49 |
| 13B | **AMLQ** | 72.30 | **79.05** | 59.92 | **79.25** | **47.35** | **67.57** |
| 70B | FP16 | 78.06 | 82.15 | 64.77 | 82.74 | 54.44 | 72.43 |
| 70B | Quip# | 77.74 | **82.43** | 64.45 | 82.28 | **54.52** | 72.28 |
| 70B | GPTQ | 77.51 | 81.72 | 64.17 | 82.70 | 53.69 | 71.95 |
| 70B | AWQ | 78.22 | 81.83 | **64.59** | 82.58 | 54.12 | 72.18 |
| 70B | **AMLQ** | **78.22** | 81.99 | 64.52 | **83.08** | 54.39 | **72.44** |

Table 2: A comparison of five downstream task % accuracy (↑) with INT4 quantization on LLaMA-2 models. **Bold** text indicates the best results. Compared with the strong baselines, AMLQ achieves the best average accuracy.

| Size | Method | MMLU | CEval (val) | GSM8K | HumanEval | Avg. accuracy (↑) |
|------|--------|------|-------------|-------|-----------|-------------------|
| 14B | GPTQ | 62.46 | 69.43 | 61.41 | 40.24 | 58.38 |
| 14B | AWQ | **64.56** | 69.09 | 63.22 | 52.43 | 62.32 |
| 14B | **AMLQ** | 64.21 | **70.49** | **64.21** | **53.61** | **63.13** |
| 72B | GPTQ | 72.82 | **79.01** | 75.95 | 59.87 | 71.91 |
| 72B | AWQ | 73.23 | 78.74 | 75.36 | 62.13 | 72.38 |
| 72B | **AMLQ** | **74.00** | 78.5 | **77.33** | **62.19** | **73.00** |

Table 3: A comparison of downstream task % accuracy (↑) with INT4 quantization on Qwen-Chat. AMLQ also achieves the best accuracy among all Qwen models.

Table 2 shows the performance of AMLQ, GPTQ, AWQ and Quip# on the zero-shot downstream tasks, which are evaluated using the *lm-eval-harness*[1]. We can observe that AMLQ achieves the most favorable results across all five tasks. The comparative analysis between AMLQ, GPTQ ans AWQ on the Qwen model is further illustrated in Table 3. The open-source *qwen-gptq-int4* is employed as a benchmark. AMLQ also shows a significant accuracy improvement over GPTQ.

## 4.3 Ablation Studies

### 4.3.1 Adaptation of the low-rank quantization error reconstruction.

Our proposed error reconstruction based on low-rank quantization stands as orthogonal to the existing quantization techniques, illustrating its potential as an integrable enhancement module for various methods. Table 4 delineates the Δperplexity measurements of the model on the WikiText-2 dataset at low bits quantization(INT2/3). The values of Δperplexity represent the relative reduction in perplexity. The low-rank error reconstruction method marginally enhances all methods at 3-bit quantization. Moreover, it significantly ameliorates the performance of OmniQuant at 2-bit quantization, and even achieves a modest improvement on the Quip# method. It demonstrates the effectiveness of the low-rank error reconstruction approach in acting as a supplementary plug-in module and augmenting the capabilities of the underlying methods.

---

[1]https://github.com/EleutherAI/lm-evaluation-harness

| Method | Q Config | LLaMA | | LLaMA-2 | | | Avg. $w$ bit |
|---|---|---|---|---|---|---|---|
| | | 7b | 13b | 7b | 13b | 70b | |
| FP16 | - | 5.68 | 5.09 | 5.47 | 4.88 | 3.32 | 16 |
| QuiP#+ | INT3 | 0.01 | 0.02 | 0.00 | 0.02 | 0.00 | 3.2 |
| AWQ+ | INT3, g128 | 0.10 | 0.04 | 0.23 | 0.04 | - | 3.3 |
| OmniQuant+ | INT3, g128 | 0.01 | 0.01 | 0.02 | 0.02 | 0.03 | 3.5 |
| QuiP#+ | INT2 | 0.02 | 0.04 | 0.01 | 0.02 | 0.02 | 2.2 |
| OmniQuant+ | INT2, g128 | 0.17 | 0.27 | 0.34 | 0.26 | 0.14 | 2.5 |

Table 4: Our proposed low-rank error reconstruction is orthogonal to other advanced quantization techniques. + denotes the introduction of our low-rank error reconstruction based on the original methods. It further closes the performance gap under low-bit quantization (INT2/3) when combined with AWQ and OmniQuant. Results are WikiText-2 $\Delta$perplexity of LLaMA models.

### 4.3.2 Impact of the low-rank error quantization reconstruction

We can achieve almost arbitrary bit quantization due to our mixed-precision strategy. Table 5 reports the perplexity of the llama2-7B model at different BPW. Our model performance shows a significant degradation below 3-bit precision without the use of low-rank error reconstruction. By introducing the low-rank error reconstruction method, the loss of the model performance can be effectively reduced at low-bit precision. When the BPW is lower, the effect of the error reconstruction method is more significant.

| AMQ | AMLQ | $\Delta$PPL | $\Delta$BPW |
|---|---|---|---|
| 15.03 | 7.68 | **7.35** | 2.3→2.5 |
| 10.46 | 6.92 | **3.54** | 2.5→2.7 |
| 9.48 | 6.27 | **3.21** | 2.7→2.9 |
| 6.42 | 5.98 | 0.44 | 2.9→3.1 |
| 6.01 | 5.82 | 0.19 | 3.1→3.3 |
| 5.86 | 5.73 | 0.13 | 3.3→3.5 |
| 5.79 | 5.66 | 0.13 | 3.5→3.7 |
| 5.7 | 5.61 | 0.09 | 3.7→3.9 |
| 5.66 | 5.57 | 0.09 | 3.9→4.1 |
| 5.57 | 5.56 | 0.01 | 4.1→4.3 |

Table 5: Perplexity on WikiText-2 aross a range of average bit precision settings. The $\Delta$PPL represents the gains from our proposed low-rank error reconstruction. The $\Delta$BPW column denotes the changes in BPW after the introduction of low-rank error reconstruction. We can observe that this method achieves significant gains under the low-bit quantization precision setting.

### 4.4 Quantitative Analysis

Given these overall performance improvements, we further analyze whether performance improvements are reflected in the selection of better bit precision and rank in each layer (as shown in Figure 2). We use the $q\_proj$ layer in the llama2-13b model as an example, which demonstrates the final selection of our proposed adaptive mixed precision quantization and low-rank error reconstruction in different blocks. Similar to the model sensitivity analysis, the higher average bit precision is used to quantize the more sensitive layers in the middle part, thus reducing the quantization error. We also observe that the ultra-low ranks are chosen to reconstruct the quantization error within those blocks with the highest precision such as 28 to 36, and even some layers choose not to reconstruct the error. The reason for

this phenomenon comes from the presence of low quantization errors after the high-bit precision quantization. However, it will lead us to prefer the extreme rank (0 or 128) due to the effect of our search strategy.

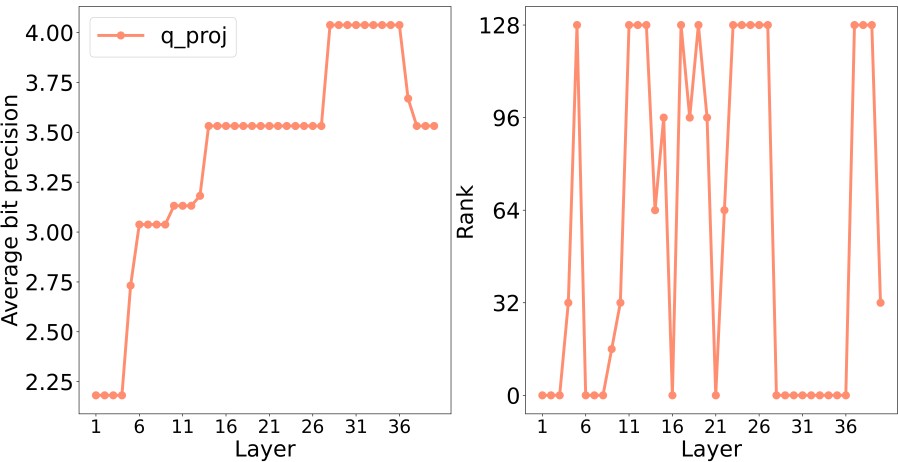

Figure 2: Quantitative analysis of AMLQ's selection of average bit precision and the rank of correction matrices in different layers.

## 5 Conclusions

In this work, we propose Adaptive Mixed Precision and Low-Rank Quantization Error Reconstruction (AMLQ), an efficient quantization method for weight-only LLM compression within low-bit or high-bit precision. AMLQ is mainly based on the observation that different layer weights have varying sensitivity to quantization bit precision. Therefore, we design an adaptive search algorithm to obtain the optimal combination of bit widths. Furthermore, the performance of most quantization techniques degrades significantly at low-bit precision and thus we introduce adaptive low-rank approximation to reconstruct quantization errors. Combining both the adaptive methods, our approach ultimately achieves the balance between performance and inference efficiency at low or high bit precision. Compared to the FP16 model, the speedup can be more than 3 times with almost no performance loss. This will further promote the practical applications of LLMs in real-world scenarios.

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
