# OpenReview forum: "Adaptive Quantization Error Reconstruction for LLMs with Mixed Precision"
_colmweb.org/COLM/2024/Conference — COLM_

### Official Review · Reviewer_cTcX · 2024-05-11

**Rating:** 7
**Confidence:** 3
**Ethics Flag:** 1

**Summary:**

Post-training quantization of large language models is a popular and generally required technique to make inference cost affordable, while avoiding the difficulties of quantization aware training. This paper proposes an adaptive mixed precision and low-rank approximation quantization error reconstruction scheme, which allows more fine-grained control over the degree of quantization applied in different layers of the model. The approach achieves state-of-the-art performance at the approximate average bit precision and good inference speed in the case of low-bit quantization, on popular benchmarks (and with suitable adaptation data).

**Questions To Authors:**

How big is the influence of the adaptation data set on the final results? I understand that the setting is standardized (i.e. same as in previous literature), but how much do results depend on the choice of fine-tuning data?

Figure 1 could do with some additional explanations (or links to appropriate definitions). What is the "normalized error sensitivity"? Does it really vary so much across the layers or is the difference mostly between the last & first vs the middle layers?

Figure 2, right plot -- what am I supposed to see here? The "rank" of the correction matrices seems to jump around quite erratically, does it really correspond to the errors?
Figure 2, left plot -- which curve in Figure 1a is this supposed to correspond to? k_proj? gate_proj?

**Reasons To Accept:**

The paper's approach is well motivated and solves a realistic problem: post training quantization is preferable to quantization aware training and the proposed approach offers more fine-grained control to meet fixed compute/ size budgets than competing approaches.

The paper is well written and the experiments seem comprehensive and well thought out. The approach seems to offer significant advances over other methods.

**Reasons To Reject:**

The set of evaluation data sets seems somewhat limited, but I am no expert in the field.

---

> ### Author Rebuttal · Authors · 2024-05-30
>
> __R1__: For comparison with other advanced methods, we utilize consistent evaluation datasets. Additionally, we apply AMLQ to multiple business datasets for text2sql and docqa tasks. The comparative results against GPTQ and AWQ are as follows:
> | task  | AMLQ | GPTQ | AWQ  | FP16 |
> |-|-|-|-|-|
> | text2sql | __88.7__ | 84.4 | 85.7 | 88.7 |
> | docqa  | __90.3__ | 86.5 | 88.2 | 91.3 |
>
> __Q1__: The quantity and diversity of a calibration dataset indeed affect the quantization results. Our experiments show that when the calibration data size exceeds 128, the performance gap in the quantized models is very small. However, as the volume of calibration data decreases, there's a notable decline in performance, a perspective also discussed in papers such as AWQ. Moreover, since our method is based on GPTQ, the optimal calibration dataset must be sampled from the entire training data, primarily impacting model generalization. From our experiments, this effect on the results is minimal. In this section, we make a fair comparison by referring to the settings from related papers.
>
> __Q2__: As described by Eq. 2, the normalized error sensitivity is defined as $\delta = \|WX-W_qX\|_2^2 / \|WX\|_2^2$. Fig. 1(a) aims to illustrate that the error sensitivity differs among weight types and layers. For the same type of weight, such as k_proj, there is a significant discrepancy between the beginning, middle, and end. Additionally, there are substantial variations between different weight types, such as k_proj and down_proj, even within the same layer. Fig. 1(b) illustrates the impact of applying quantization error reconstruction methods across various bits per weight (bpw), demonstrating significant performance improvements at lower bit widths. In Fig. 1(c), SVD represents the direct calculation of the rank of $\Delta W = W - W_q$. L^2QER scales $\Delta W$  then calculates its rank. AMLQ determines the rank of the covariance matrix of $\Delta Y$ from Eq. 7-10.
>
> __Q3__: Fig. 2 shows the search results for bits per weight (bpw) and the required rank of the error reconstruction matrix for each q_proj layer in llama2-13b, which is also related to error sensitivity. Higher sensitivity in intermediate layers necessitates higher average bpw. For example, in layers 28-36, using the highest average bpw results in smaller quantization errors, so only the minimum necessary rank is used for error reconstruction, indicating no need for additional low-rank matrices.

---

> > ### Comment · Reviewer_cTcX · 2024-06-05
> >
> > Thank you for providing feedback. It is good to see that the size of the calibration dataset does matter (as one would expect), but that the chosen size is sufficient. Thank you also for clarifying results on other datasets and the further clarifications - please consider if some could be added to increase the value of the paper to the readers. Overall, I will maintain my rating.

---

### Official Review · Reviewer_n3yk · 2024-05-11

**Rating:** 4
**Confidence:** 3
**Ethics Flag:** 1

**Summary:**

This paper investigated the post-training quantization method for LLMs. It mainly combines two methods for two observed issues: 1) given the observation that different layers have varying sensitivity to quantization, it uses adaptive mixed precision search to search for optimal precision for different layers. 2) it also uses an existing low-rank approximation LQER to reconstruct quantization errors.  Combining both of the above methods, it seems the proposed approach could achieve a balance between performance and inference efficiency, especially for low-bit settings.

**Questions To Authors:**

Q1. Not quite clear the difference between the proposed low-rank reconstruction error and the LQER. What is the benefit? Please clarify more.
Q2. Could you explain more about Table 1 (Avg bit 2.0+). why is there no LQER for the 3 and 2-bit set? Also, please show your searched optimal mixed precision for each layer and their quantization loss. What can be improved?

**Reasons To Accept:**

The proposed approach of combining two adaptive methods seems useful and could balance the performance and inference efficiency by considering both per-layer quantization and low-rank.

**Reasons To Reject:**

1. The independent assumption and the sum of quantization error seem problematic as a good approximation.
2. Equation (6), not sure the group setting is the same as the claimed "layer-wise" quantization bit-width. Not sure it is the optimal dynamic setting.

---

> ### Author Rebuttal · Authors · 2024-05-30
>
> __R1__: Your perspective is highly valuable. We've strategically designed AMLQ to minimize the total quantization error across linear layers while keeping the overall average bits per weight (bpw) under a predefined target. Through experimental observation, we identify that different layers have varying sensitivities to quantization. Accordingly, we tailor bit allocation to minimize total quantization errors—assigning lower bpw values to shallower, less sensitive layers, and higher bpw values to deeper, more sensitive ones.
>
> __R2__: We apologize for any confusion. Group size, mixed precision strategy, and proportions are critical for determining the layer's bit width for quantization. For example, dividing a (5120,5120) weight layer into 128-sized groups yields 40 groups. Quantizing 40% at 2 bits, 30% at 3 bits, and the rest at 4 bits averages to 3.3 bpw. The smaller groups require more quantization parameters, leading to a larger bit width. We select group sizes as multiples of 32 to optimize CUDA kernel efficiency, typically requiring smaller groups for lower precision but greater proportions. Therefore, we set three group sizes (32,64, and 128) to match the mixed precision quantization. Tables 1 and 2 validate our method.
>
> __Q1__: Both our method and LQER use two low-rank matrices to approximate a decomposed matrix, but while LQER decomposes the quantization error matrix $\delta_W$, we decompose $\delta_Y$'s covariance matrix (in Equations 10). Lower matrix rank means less error in approximation, and our experiments show that activation values are low-rank, meaning the rank of $\delta_Y$ is lower compared to the quantization error matrix $\delta_W$. Fig.1(c) in our paper visualizes this low rank, supporting our method's superior error reconstruction.
>
> __Q2__: Table 1 compares the perplexities on WikiText-2 of AMLQ with other strong baselines, using results from their original papers. LQER has no results below 4-bit and its closed-source code prevents reproduction for 2-3 bit results. Figure 2 shows lower bit widths for shallower layers and higher for deeper ones in the llama2-13b model, aligning with k_proj’s quantization error trends in Figure 1(a). Our greedy binary search might not find the best quantization combination, but better strategies and layer-wise fine-tuning could improve results. We will optimize the search strategy for a balance between effectiveness and efficiency. Overall, AMLQ performs better in most cases.

---

> > ### Comment · Reviewer_n3yk · 2024-06-03
> > **Thanks for the rebuttal**
> >
> > Thanks for the clarification. Overall, the varying sensitivities of each layer are well-known observations. The motivation for adapting mixed precision and activation-based low-rank reconstruction for each layer is also straightforward. Given the observation and problem is not new, I would like to see a sound and exciting solution.  However, it seems the solution is preliminary and working in progress. I still think it is not good enough.

---

> > > ### Author Response · Authors · 2024-06-04
> > > **Thanks for your feedback**
> > >
> > > Thank you for your feedback. While we acknowledge that the underlying motivations behind employing mixed precision and activation-based low-rank reconstruction are indeed intuitive and address known observations, we emphasize that our method is driven by a concrete problem-solving approach. This stems from real-world challenges, guiding our development of the proposed solutions which have shown significant performance improvements and efficiency gains in practical applications.
> > >
> > > We understand your concerns regarding the current stage of our solution being perceived as preliminary. We are committed to continuously refining our methodology, focusing on rigorous theoretical analysis and practical application enhancements. Our future work will aim at not only solidifying the theoretical underpinnings but also at demonstrating the utility and robustness of our approach in diverse real-world scenarios.
> > >
> > > We truly appreciate your suggestions and the opportunity to further develop and optimize our work. Your feedback is invaluable in guiding our efforts to achieve a solution that is both theoretically sound and practically effective.

---

### Official Review · Reviewer_SfE9 · 2024-05-13

**Rating:** 4
**Confidence:** 4
**Ethics Flag:** 1

**Summary:**

This paper proposed a rank adaptation and mixed-precision method for improved accuracy of quantized adapter-based LLM inference. The proposed method was motivated by the varying quantization sensitivity of the Transformer layers to assign higher bit-precision and/or ranks to the more sensitive layers. The authors proposed a greedy method to optimize the bit-precision and rank allocation, and justify their choice by comparing quantization performance with the other quantization methods.

**Questions To Authors:**

The explanation about Fig. 1 seems to lack explanation details. For example, how the authors obtain Fig, 1(C)? Which model or calibration dataset are used for this analysis?

**Reasons To Accept:**

- The authors provide a practical method to robustify extremely-low precision quantized LLM inference.
- The authors provide an extensive performance comparison with the existing state-of-the-art quantization methods.

**Reasons To Reject:**

- The proposed methods lack theoretical justification of why it overcomes the challenges of the previous method. In particular, the authors claimed that the output activation values are low-rank, but there is no explanation why they reached to this claim. If it is a simple observation, the readers would expect more extensive empirical study to support this claim.

- Also, the proposed methods seem to be composition of well-known quantization methods. The proposed mixed precision allocation method seems to be a straightforward application of existing mixed precision methods into LLMs. Also, the adapter reconstruction (also called adapter initialization) is widely studied; e.g.,

> LoftQ: LoRA-Fine-Tuning-Aware Quantization for Large Language Models (https://arxiv.org/abs/2310.08659)

> LQ-LoRA: Low-rank Plus Quantized Matrix Decomposition for Efficient Language Model Finetuning (https://arxiv.org/abs/2310.08659)

> ApiQ: Finetuning of 2-Bit Quantized Large Language Model (https://arxiv.org/abs/2402.05147)

Due to lack of in-depth analysis of the proposed method in comparison with these existing works, the novelty of the proposed methods is not clear.

- Also, the experimental results show that the proposed method is only marginally better than QuiP#; even in Table 4, the accuracy gain from the proposed rank adaption for QuIP# seems to be very marginal. Therefore, it is not clear if the proposed method is empirically superior.

- The proposed optimization technique seems to be greedy, and there is no analysis of its optimality.

- The authors claimed that the proposed methods outperform the other methods thanks to its efficient CUDA implementation. However, there is no code provided or experimental results comparing the inference speed.

---

> ### Author Rebuttal · Authors · 2024-05-30
>
> __R1__: Our paper addresses two key issues with prior quantization methods. First, we tackle varying sensitivity to quantization across different layers and channels by proposing an adaptive mixed-precision approach. Our method enables a theoretically superior combination of precisions, proven to outperform uniform precision techniques. Second, we counter the significant accuracy drop in low-bit conditions witnessed in previous methods with our novel low-rank quantization error reconstruction. Through theoretical analysis and empirical evidence, we show how output activations are low-rank and how they can be leveraged to mitigate quantization errors. This validates the effectiveness and theoretical soundness of our approach.
>
> __R2__: Our method innovates for LLMs by capitalizing on uneven quantization sensitivity across LLM layers and the advantageous activation value reordering strategy, similar to that in GPTQ. The adoption of low-rank matrix error reconstruction improves quantization, particularly at ultra-low bits. Our low-rank assumption leads to a unique matrix initialization that requires minimal data, divergent from backpropagation-reliant methods, and is especially effective post-training, tailored for LLMs.
>
> __R3__: Our AMLQ method offers time and performance superiority to QuiP#, which succumbs to long quantization times and complex design. Despite marginal improvements QuiP# may demonstrate due to its unique codebook design, AMLQ surpasses conventional methods like GPTQ, AWQ, and OmniQuant, showcasing its broad empirical superiority.
>
> __R4__: Facing a large search space, we adopt a relative optimality approach with our iterative, bisecting greedy strategy. This balances optimality and computational efficiency by systematically narrowing down the layer combinations, guiding us towards a practical solution within the target bits-per-weight.
>
> __R5__: We will soon provide a comprehensive speed analysis for the Llama2-13B-chat model, with preliminary tests highlighting our solution's inference speed superiority. At 2.5 bits, our method is significantly faster than QuiP# at 2 bits and a non-quantized model at fp16, achieving 46.08 tokens/s, which further increases to 117.7 tokens/s with our specialized inference framework.
>
> __Q1__: The experiments in Fig. 1 are carried out in llama2-13b. The calibration dataset is sampled from the WikiText2. Due to space limitations, please refer to the detailed explanation of Fig. 1 in Reviewer 4's Q2.

---

> > ### Comment · Reviewer_SfE9 · 2024-06-03
> > **Comments after Rebuttal**
> >
> > I thank the authors for their responses, but the responses they provided did not sufficiently address my concerns. Above all, the authors claimed in the rebuttal that the proposed idea is "proven" to outperform the existing techniques with "theoretical analysis," but it is hard to understand the theoretical reasoning behind their methods. The authors admitted that their methodology is greedy, but they did not provide the ablation study to empirically support the superiority of their greedy method. The authors also share a plan to provide a comprehensive speed analysis, but it does not seem to be included in this rebuttal. Therefore, I would like to maintain my original score.

---

> > > ### Author Response · Authors · 2024-06-03
> > > **Additional Notes on Inference Speed**
> > >
> > > In our evaluation conducted on a single A100 GPU, we tested the Llama2-13B-chat model's inference speed. For the Quip# method, I used the evaluation code available at https://github.com/Cornell-RelaxML/quip-sharp, with a batch size of 1. The inference speed for the 2-bit quantized model was measured at 21.2 tokens/s, whereas the non-quantized fp16 model operated at 27.02 tokens/s. Clearly, quantization via Quip# results in a reduction of inference speed compared to the non-quantized model.
> > > Maintaining the same experimental setup, our model, being unable to achieve a full 2-bit quantization, was assessed using the closest approximation of a 2.5-bit quantized model. We tested this by only replacing the quantized linear layers. Our approach yielded an inference speed of 46.08 tokens/s. Notably, when paired with our internally developed inference framework, we were able to reach an impressive 117.7 tokens/s, far surpassing the non-quantized model's speed of 45.3 tokens/s.

---

> > > ### Author Response · Authors · 2024-06-03
> > > **Explanation of Greedy Search Strategy**
> > >
> > > Thank you for your comments. Due to the rebuttal word limit of 2500 characters, we did not provide a detailed explanation of our theory. We will further elaborate on it.
> > >
> > > We would like to emphasize that, unlike using the same quantization precision for each linear layer, our mixed-precision quantization approach allows different quantization precisions to be applied to different linear layers and integrates them. This method ensures minimal overall error while meeting the given bits-per-weight (bpw) requirement. For instance, when the mixed-precision target bpw is set to 4, applying 4-bit precision quantization to all linear layers adheres to the bpw condition but may not result in the minimal overall error combination. Given the vastness of the search space, finding the optimal solution within a limited time frame is challenging. Therefore, we employed a greedy strategy to obtain a relatively optimal solution.
> > >
> > > This is essentially a binary greedy strategy where we minimize the maximum quantization error $N_{max}$ under the  target bpw condition. $N_{max}$  is defined as the largest quantization error obtained when applying the target bpw to each linear layer. $N_{max}$ serves as the upper bound for our binary search, with the starting point being the precision quantization of all linear layers using the same precision. In detail，We use $N_{max}/2$ as our target value during the binary search. In this process, we eliminate all combinations with a quantization error exceeding  $N_{max}/2$ at each layer. Then, for the remaining combinations that exhibit quantization errors below this threshold, we select the combination with the smallest bits-per-weight (bpw) to compute the model's overall bpw value.
> > >
> > > If this computed bpw is less than our target, it suggests the existence of a viable solution, although this solution may not minimize the quantization error. We continue to binary search using smaller target values, repeating the process until we either cannot find a combination that satisfies the target bpw, or the precision of our binary search for the quantization error target reaches below 1e-5. At that point, we terminate the search and adopt the last satisfying solution as our final resolution.

---

> > > ### Author Response · Authors · 2024-06-04
> > >
> > > Thank you for your feedback. We acknowledge the concerns you have raised about the theoretical basis and empirical support for our greedy methodology. However, our method maintains good performance and inference efficiency in practical applications, which is also a manifestation of application innovation. In response:
> > >
> > > - Theoretical Clarification: We will enhance our theoretical analysis to better elucidate the method's efficacy, aiming for clearer comprehension and acceptance.
> > >
> > > - Empirical Evidence: We plan to conduct an ablation study to provide empirical evidence for the superiority of our method, specifically demonstrating its performance and efficiency advantages over existing techniques.
> > >
> > > We appreciate your insights and agree on the necessity of reinforcing the theoretical reasoning and empirical justification. Our team is committed to addressing these issues and improving the manuscript's clarity and rigor. Thank you for allowing us the opportunity to refine our work. We look forward to your reconsideration.

---

### Official Review · Reviewer_SayR · 2024-05-13

**Rating:** 6
**Confidence:** 4
**Ethics Flag:** 1

**Summary:**

This paper present Adaptive Mixed Precision and Low-Rank Quantization Error Reconstruction (AMLQ), a quantization method for LLMs that adapts bit precision based on weight sensitivity. Different from previous works, it might be the first to indicate the sensitivity of different layers. This approach balances performance and efficiency, achieving speedups with minimal performance loss, enhancing the practical application of LLMs.

**Questions To Authors:**

The two innovations proposed in this paper AMLQ are meaningful, and the ablation studies in the paper confirm this; however, the experimental results in Table 1 are not significant overall. The method in this article follows the GPTQ paradigm, yet this classic and relatively old work seems to struggle with quantization scenarios below 4 bits. It is recommended that the authors explore settings on extremely low-bit quantization and consider how to apply your ideas in this setting, which might yield more significant results.

**Reasons To Accept:**

This paper notes that different layers have varying sensitivities to quantization and introduces the concept of quantization error reconstruction along with its theoretical basis.

**Reasons To Reject:**

Although AMLQ's quantization error reconstruction shows some effectiveness compared to GPTQ and AWQ, its performance is not significant compared to the current strongest baseline, and in most cases, it does not have a competitive advantage.

---

> ### Author Rebuttal · Authors · 2024-05-30
>
> __R1__: The primary advantage of our proposed AMLQ lies in its ability to maintain quantization effectiveness and inferencing speed under ultra-low quantization bits while achieving SOTA average scores on downstream tasks. As seen in Table 1, AMLQ significantly outperforms most existing quantization approaches, with only QUIP# ranking higher. However, QUIP# employs layer-wise fine-tuning and an E8 codebook. AMLQ can save 10x quantization time and increase inference speed 3x compared to QUIP#. In Table 2, AMLQ achieves SOTA overall scores across five downstream tasks, demonstrating its superior performance in practical deployment applications.
>
> __Q1__: __(1)__ In the public evaluation datasets presented in Table 1, most existing quantization methods perform well at 4 bits, with AMLQ ranking among the top. Notably, OmniQuant and QUIP#, which show comparable effects to AMLQ, require backpropagation to update parameters during quantization, making AMLQ more advantageous in terms of quantization time and computing resource utilization. Furthermore, our AMLQ exhibits a significant performance advantage under 2-3 low-bit quantization (shown in Table 1). \
> __(2)__ We wholeheartedly agree with your perspective. As you mentioned, the GPTQ method is an excellent work from an earlier time, yet it underperforms in ultra-low bit quantization scenarios. In contrast, AMLQ excels in ultra-low bit quantization applications. In Table 1, we showcase the quantization performance at 2-3 bits across various datasets. AMLQ achieve substantial improvements over existing strong baseline models and performed comparably to the QUIP# method. It’s worth noting that QUIP# utilizes layer-wise fine-tuning and an E8 codebook, whereas AMLQ can save 10x quantization time and improve 3x inference speed compared to QUIP#. Furthermore, according to experimental results from the QUIP# paper, AMLQ's performance surpasses that of QUIP#'s non-fine-tuned version in most scenarios.

---

### Decision · Program_Chairs · 2024-07-10

**Decision:**

Accept

**Comment:**

Quality
- Pro
  - Comprehensive and well-thought-out experiments
  - address important problem in PTQ
- Con
  - Some concerns of empirical quality in particular with respect to low-rank activation experiments

Clarity
- Pro
  - well-written
- Con
  - some figures unclear
  - additional theoretical justification would add to clarity

Originality
- Pro
  - novel way of fine-grained quantization by mixed precision + low-rank approximation
- Con
  - techniques are a composition of existing techniques
  - lacks theoretical insights why it is working

Significance
- Pro
  - More fine-grained control of PTQ
  - provides new insights into PTQ
- Con
  - marginal improvements over existing methods

**Decision**

The paper is well-motivated and well-written, and it provides a novel way of fine-grained quantization. The overall experimental quality and rigor is sufficient. The main points are standing against acceptance is the lack of theoretical insight, the presented method being just a combination of existing methods, and the overall marginal improvements compared to previous work. Overall, the reviewers deem the work borderline and falls slightly short of the COLM acceptance curve. As such, I do not recommend acceptance.

EDIT: I chatted with the PCs about this case and was encouraged to include my personal opinion on this matter to reach a decision.

I think this paper is a case of work that does not change the field, but augments existing evidence between ideas and helps researchers to really figure out what other papers might have strong signals for future research. I think these papers are essential for a healthy research environment. I, personally, am all for accepting this paper.

With this, I update my decision to recommend "Maybe acceptance", to balance the judgement of the reviewers with my personal opinion.